# Short-Term Changes in Erosion Dynamics and Quality of Soils Affected by a Wildfire and Mulched with Straw in a Mediterranean Forest

**Manuel Esteban Lucas-Borja** [1], **Misagh Parhizkar** [2] **and Demetrio Antonio Zema** [3,*]

1   Escuela Técnica Superior Ingenieros Agrónomos y Montes, Campus Universitario, Universidad de Castilla-La Mancha, E-02071 Albacete, Spain; manuelesteban.lucas@uclm.es

2   Soil Science Department, Faculty of Agricultural Sciences, University of Guilan, Rasht 41996-13776, Iran; misagh.parhizkar@gmail.com

3   Department AGRARIA, Mediterranean University of Reggio Calabria, Loc. Feo di Vito, I-89122 Reggio Calabria, Italy

*   Correspondence: dzema@unirc.it

**Abstract:** Wildfire heavily impacts the quality of forest soils, and the precipitation occurring immediately after fire can determine high runoff and erosion rates, which may lead to noticeable soil degradation. Mulching is commonly used to limit the hydrological impacts of wildfire and climate, but this post-fire management technique may alter the erosion–deposition dynamics at the hillslope scale and, consequently, alter soil quality. In order to explore the magnitude and significance of these changes (little was studied in the literature until now), this communication reports the first results of a field activity that evaluated the changes in soil quality in areas affected by a wildfire and subjected to different post-fire treatments in Mediterranean forests. The main properties of sediments eroded from burned and untreated soils, and mulched soils (using a straw dose of 0.2 kg/m$^2$ of dry weight), were measured after the first rainstorm (height of 37 mm and maximum intensity of 11.6 mm h$^{-1}$) occurring two months after a wildfire (occurred on 30 June 2016) in a pine forest of Castilla-La Mancha (Spain). This event produced a runoff volume of 0.07 ± 0.02 mm in mulched soils and 0.10 ± 0.10 mm in non-mulched soils; soil loss was 0.20 ± 0.06 g/m$^2$ in the mulched area and 0.60 ± 0.60 g/m$^2$ in the non-mulched area. In comparison to burned and non-treated areas, this study showed: (i) increases in salinity, and reductions in organic matter, nutrients, nitrates, and micro-elements in burned and untreated soils; (ii) reductions in runoff (−20%) and in soil erosion (−60%) as a result of mulch cover; (iii) effectiveness of mulching in limiting the declines in soil quality detected in burned and eroded areas; and (iv) transport of low amounts (less than 10–15%) of some compounds (organic matter and nutrients) downstream of the fire-affected areas (both mulched and untreated). Phosphorous runoff toward valley areas and nitrate incorporation into the soil, detected in both mulched and untreated areas, require attention, since these processes may cause eutrophication of water bodies or nitrate pollution in groundwater.

**Keywords:** wildfire effects; soil loss; deposition; organic matter; nutrients; micro-elements; pine forest; post-fire treatments

## 1. Introduction

Forest soils have generally good quality and functionality, due to their intrinsic properties (e.g., high contents of organic matter and nutrients, aggregate stability, porosity, and stable microbial activity) [1–3]. These characteristics let forest soils produce an important ensemble of ecosystem services, such as oxygen production, carbon storage, regulation of surface water and energy fluxes, and support of biodiversity [4,5]. However, the quality and functionality of forest soils is threatened by some natural and anthropogenic causes (e.g., extreme weather events and fires of different intensities) [2,6], which are reasons

for soil degradation and plant destruction with heavy damage not only for forest areas, but also for urban and peri-urban zones [7,8]. Wildfires strongly modify the physical, chemical, and biological properties of forest soils and completely remove vegetation [9–12], and these fire actions determine often-irreversible damage to forest soils or degradation rates that need several years or even decades to be restored [10,13]. Moreover, runoff and erosion heavily increase immediately after a wildfire, with slow decreases (several months or some years), and they tend to decrease with time [14–16]. These negative impacts on the forest environment can be aggravated in the Mediterranean Basin [17], where the wildfire occurrence and effects are generally more severe compared to other ecosystems, due to the intrinsic climatic characteristics (dry and hot summers and frequent and intense rainstorms in autumn, immediately after the wildfire season) [10,11,18,19]. Furthermore, these climatic conditions are expected to increase the wildfire frequency and burned area by the forecasted climate scenarios [20,21]. For instance, in Spain, forests are severely affected by wildfires in summer, and, in the last 10 years, more than 3000 km$^2$ of forests have burned [22,23].

Post-fire management techniques, both at hillslope and catchment scales, are targeted to reduce as much as possible many negative impacts of wildfire, and their effectiveness in reducing runoff and erosion rates in fire-affected soils have been demonstrated [24,25]. However, post-fire management heavily changes the properties of burned soils [26]. Soil mulching is one of the common post-fire management techniques [16,25,27], especially with the use of straw or forest residues [28,29]. The mulch layer increases ground cover (limiting rainsplash erosion) and water infiltration (decreasing surface runoff) and, in general, quality of soil [30]. However, some negative effects of mulching have been reported in the literature, such as a reduction in unsaturated hydraulic conductivity [18], displacement by wind, or possibility of introducing diseases and parasites in forests [31,32].

Ample literature is available about the effectiveness of soil mulching in reducing the surface runoff and erosion rates in fire-affected areas in several environmental conditions (e.g., [33–39]). Less attention has been paid to post-fire changes in soil quality due to the erosion–deposition dynamics at hillslope scale. For instance, losses of organic matter have been little studied, despite their importance in soil fertility and plant regeneration after fire [40]. These changes in soil properties occurring after fire and soil treatments are an important research issue, since erosion may contribute to soil depletion in some elements or compounds (organic matter, nutrients, or microelements), which are essential to support vegetation regeneration in areas burned by high-severity fires. However, in spite of the beneficial effects of mulching, further research should explore these effects more in depth, especially in areas with severe risks of soil erosion and degradation, and in the short-term after wildfires [16]. Evaluating the changes in the properties of the soil (mulched or not subject to any post-fire management techniques) eroded in the early stage of the "window of disturbance" that occurs immediately after wildfire [41] is essential to understand the quality loss due to fire and the effectiveness of mulching in limiting this loss. In this window of disturbance, the first post-fire rainstorms increase the sediment yields, determining a peak over the background levels of erosion, because soil is most vulnerable to detachment [10]. The first event, in particular, is responsible for the main changes in soil properties and the highest erosion (e.g., [27,30,42]).

To our best knowledge, a similar evaluation has not been carried out until now, or only partial results are available. Prats et al. [39,41] have evaluated the changes in organic matter after polyacrylamide application and forest-residue mulching for reducing post-fire runoff and soil erosion in eucalypt stands of Portugal. In the same environment, Lopes et al. [42] investigated the effects of ploughing and mulching on soil and organic matter losses. In Spain, Francos et al. [26] measured a large dataset of soil properties after post-wildfire management techniques (Cut and Remove, Cut and Leave, and No Treatment), and Lucas-Borja et al. [2] analyzed the effects of wildfire and logging on soil functionality in the short-term. However, these authors did not consider the effects of mulching.

To fill this gap, this communication proposes the first results of an ongoing and wide research activity that aims at evaluating the changes in soil quality in areas affected by a wildfire and subject to different post-fire treatments in Mediterranean forests. More specifically, this preliminary contribution focuses on the changes measured in the main properties of sediments eroded from burned and untreated soils, and mulched soils after the first rainstorm occurring after a wildfire in pine forest of Castilla-La Mancha (Spain). We hypothesize that mulching is effective in limiting the loss of organic matter, nutrient, and microelement contents compared to untreated soils.

## 2. Materials and Methods

### 2.1. Study Area

The experimental investigation was carried out in the Sierra de las Quebradas forest (geographic coordinates: 38.5164048N, −1.8318104E, close to Liétor, Castilla-La Mancha, Spain; Figure 1a). The elevation ranges between 520 and 770 m a.s.l. and its aspect is W-SW. The climate of the area is Mediterranean semi-arid (BSk, Köppen-Geiger classification, [43]) with mean annual rainfall and temperature of 282 mm and 16 °C, respectively. According to the USDA taxonomy [44], the soils are *Inceptisols* and *Aridisols* with sandy-loam texture.

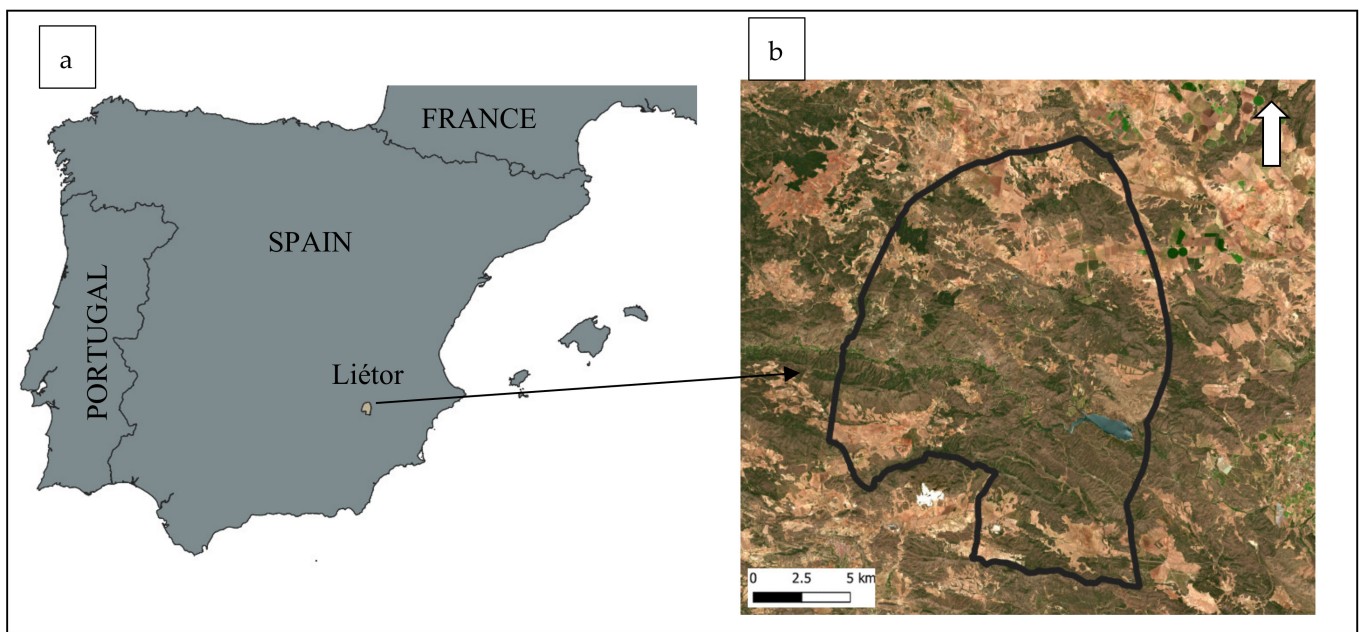

**Figure 1.** Geographical location (**a**) and aerial map (**b**) of the experimental site (Liétor, Castilla-La Mancha, Spain; geographical coordinates: 38.5164048 N, −1.8318104 E).

Logging was the main disturbance of the forest stands. Forest management was practiced supporting wood productivity. Due to progressive abandonment and the reforestation works by the forest authorities, the landscape consisted of *Pinus halepensis M.* of natural origin, and *Quercus cocciferae*. Tree (*Pinus halepensis M.*) cover of the studied area had a mean density between 500 and 650 trees ha$^{-1}$ and a height between 7 and 14 m. The main species of shrubs and herbs of the forest were *Rosmarinus officinalis* L., *Brachypodium retusum* (Pers.) Beauv., *Cistus clusii* Dunal, *Lavandula latifolia* Medik., *Thymus vulgaris* L., *Helichrysum stoechas* (L.), *Stipa tenacissima* (L.), *Quercus coccifera* L., and *Plantago albicans* L.

### 2.2. Experimental Design

In July 2016, the forestland (about 800 ha) was burned by a crown wildfire with a tree mortality of 100%. Immediately after the fire, a forest area of about 5 ha was identified for this study (Figure 1b). Immediately after the wildfire, 12 hydraulically isolated plots measuring 20 m$^2$ each were set up in the selected wildfire-affected area. All

plots were very similar regarding both soil and forest characteristics in terms of slope, aspect, and vegetation type. The aspect was north and the slope was 30–35% for all plots. The vegetation type before the wildfire was the same for all plots (see Lucas-Borja et al., 2020a, for more details). All the experimental plots were characterized as burned with high severity by the forest service of Castilla-La Mancha. In September 2016, mulch was manually applied to six of the previously selected plots at a straw dose of 0.2 kg/m$^2$ (dry weight). This dose was setup according to the indications by Vega et al. (2014) for Northern Spain, where a soil cover higher than 80% was achieved. Mulched plots were randomly selected. The mulch, composed of straw, covered 95% of the total area, and its thickness was 3 cm.

### 2.3. Data Collection

### 2.3.1. Hydrological Observations

For the first rainstorm (21 October 2016), precipitation amount and intensity were measured using a rain-gauging station (WatchDog 2000 Series model) located in the burned area. The runoff volume was measured in 12 tanks (each one 100 L) installed downstream of the 20 m$^2$ hydraulically isolated plots (6 burnt and mulched plots and 6 burnt and non-mulched plots). Samples (each one 0.5 L) of runoff water were collected after manual shaking, and sediment concentration was measured in the laboratory (Lucas-Borja et al., 2019); soil loss was calculated as the product of runoff volume by sediment concentration. Moreover, the eroded soil deposited in the 12 sediment fences set up at the bottom of each plot was manually collected and then weighed in the field. Overall, 12 runoff samples collected in tanks, 12 soil samples collected in the sediment fences, and 12 soil samples collected at the surface plots were analyzed. For soil samples, 500 g composite samples obtained both at the sediment fences and at the plots surface were oven-dried (at 105 °C) for 24 h in the laboratory.

### 2.3.2. Soil Characterization

The pH was measured in a 1/5 ($w/v$) aqueous extract using a pH meter. Texture was analyzed using the method of Guitián and Carballas [45]. Nitrates (N-NO$^{3-}$) were determined using the method described by Keeney and Nelson [46]. Organic matter (OM) content was measured according to Nelson and Sommers [47]. The C/N ratio was calculated as reported by Lucas-Borja et al. [48]. The total contents of N, P, K, Na, Mg, and Ca were determined, after nitric-perchloric acid digestion, by ICP spectrometry. The concentrations of sulphate in the water extract (1:10, soil:water) were analyzed by HPLC using a conductivity detector. The water extract was obtained by shaking for two hours a mixture of soil and distilled water (1:10 soil:water ratio), centrifuging, and filtering. The electrical conductivity was measured directly using a meter which reads directly in conductivity values. Cation exchange capacity (CEC) was calculated according to Barker et al. [49].

### 2.4. Statistical Analysis

First, a 2-way analysis of variance (ANOVA) was applied to all soil parameters (considered as response variables), assuming as factors the soil condition ("non-mulched" and "mulched") and erosion dynamics ("eroded sediments" and "deposited sediment"). The statistical significance of the differences in the response variables was evaluated through the pairwise comparisons using Tukey's test (at $p < 0.05$). To satisfy the equality of variance and normal distribution of soil sample distribution, the data were processed by normality tests or were square-root-transformed whenever necessary.

Second, a principal component analysis (PCA) was applied to the soil samples, in order to identify the derivative variables (principal components—PCs) and simplify the analysis of the several soil properties among soil conditions and erosion dynamics.

Finally, the soil samples were grouped in clusters using agglomerative hierarchical cluster analysis (AHCA), a distribution-free ordination technique to group samples with

similar characteristics by considering an original group of variables. The Euclidean distance was used as a similarity–dissimilarity measure [50].

The statistical analysis was carried out using the XLSTAT release 2019 software.

## 3. Results

The first event that occurred after the wildfire was a rainstorm with a height of 37 mm and a duration of about 9 h, which led to a mean and maximum intensity of 4.11 and 11.63 mm/h, respectively. This event produced a runoff volume of $0.07 \pm 0.02$ mm in mulched soils and $0.10 \pm 0.10$ mm in non-mulched soils; soil loss was $0.20 \pm 0.06$ g/m$^2$ ($2 \times 10^{-3} \pm 0.6 \times 10^{-3}$ tons/ha) in the mulched area and $0.60 \pm 0.60$ g/m$^2$ ($6 \times 10^{-4} \pm 6 \times 10^{-3}$ tons/ha) in the non-mulched area (Figure 2).

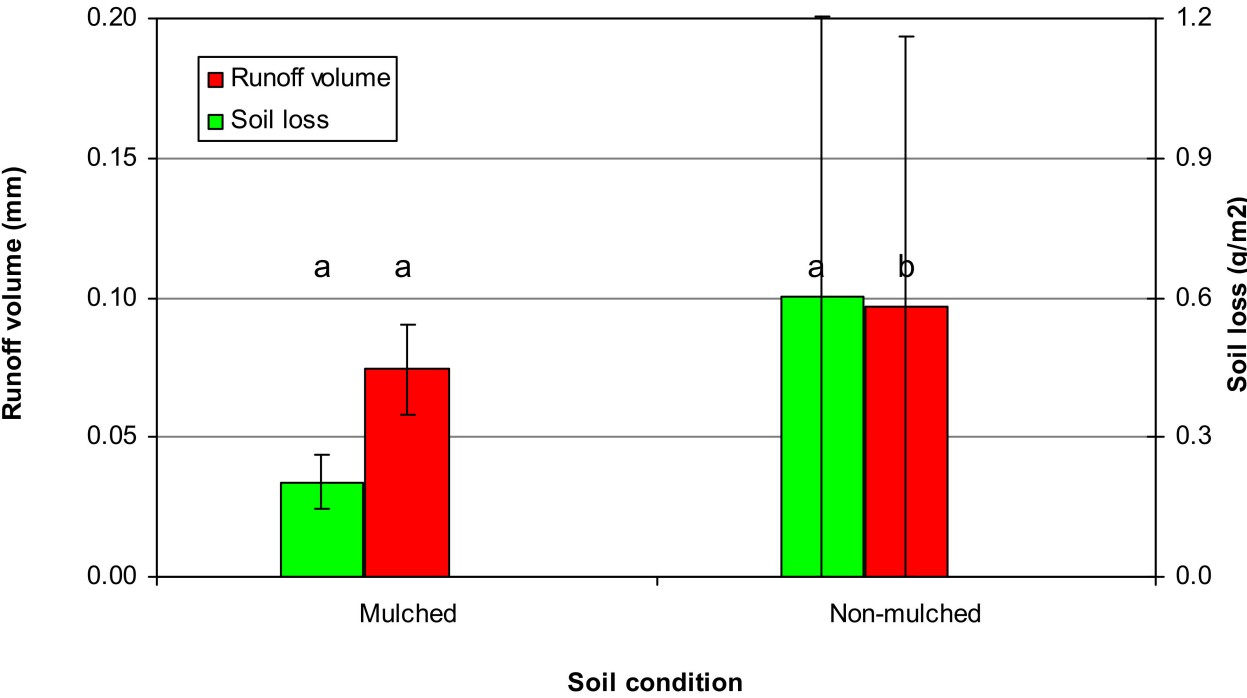

**Figure 2.** Surface runoff volume and soil loss after the first precipitation in soils affected by a wildfire and treated with straw mulch in Liétor (Castilla-La Mancha, Spain). Different letters indicate significant differences after Tukey's test ($p < 0.05$).

The ANOVA showed that all the changes in the analyzed parameters of mulched vs. non-mulched soils were significant ($p < 0.05$), except for Mg. Moreover, the comparison of these parameters between deposits highlighted that the differences in silt, pH, Na, and CEC between the eroded vs. deposited sediments were not significant, while the changes in the other properties were always significant (Appendix A and Abbreviations).

In more detail, the analysis of the samples of the mulched and non-mulched soils (surface layer) highlighted that (Figure 3a):

- Mulching increased the sand fraction (+7.8%), while the volume of clay particles significantly decreased (−25%);
- The pH practically did not vary (+0.5%), while, in contrast, EC significantly increased in the mulched soils (+11.6%);
- OM and nutrients significantly increased in mulched soils (OM +28.2%, N 26%, P 73%, and K +38.9%), and these variations determined an increase in C/N (+6.6%);
- The cation contents slightly varied in mulched soils compared to the unburned and untreated areas (maximum variation of −11.7% detected for Na);
- Nitrates noticeably depleted in mulched soils (−80.9%), while a high increase (+144%) in sulphate content was detected;
- CEC of the mulched soils showed an increase (+15.6%).

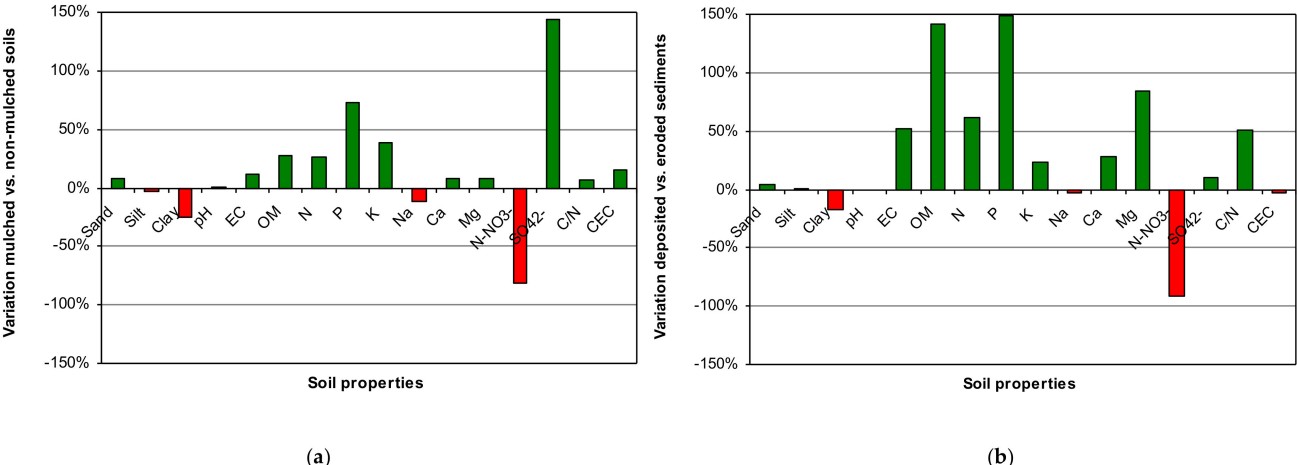

**Figure 3.** Percent changes in mulched vs. non-mulched plots (**a**) and eroded vs. deposited sediments (**b**) of the surface layer of soils affected by a wildfire and treated with straw mulch in Liétor (Castilla-La Mancha, Spain). Bars express the average values.

Moreover, the comparison of properties of the surface layer between the eroded soil (remaining on the hillslope after the rainy event) and deposited soil (in sediment fences) showed (Figure 3b):

- Stable silt and sand contents, and a significant decrease in clay content in the eroded fraction (−17.3%);
- A slight and non-significant variation (−0.5%) in soil pH;
- A significant increase (+51.7%) in EC of the deposited sediments;
- Noticeable increases in OM and nutrients (OM +141%, N +61.4%, P 149%, and K +22.7%) in the deposited sediments, which furthermore let C/N rise (+50.9%);
- An enrichment in Ca (+28.4%) and Mg (+84.6%) in the deposited fraction, and a slight variation (−3.3%) in the Na content;
- A depletion of nitrates (−91.8%) in the deposited sediments;
- Stability in CEC (−2.6%).

PCA identified two derivative variables (PC1 and PC2), which together explained 74% of the total variance, of which 56.1% was reflected in the first component. Among the original variables, the contents of OM, all nutrients, Ca, Mg, EC, and C/N had high (>0.75) and positive loadings on the PC1, while the loading of clay and nitrates were negative, although high also in this case (>0.80). Only the sand and silt contents also had high loadings (−0.71, +0.70, and 0.76, respectively) on the second PC (Figure 4a).

The AHCA allowed clustering the soil samples according to the soil condition and forest-erosion dynamics (Figure 5). More specifically, immediately after fire, three similar clusters of soil samples were evident: (i) mulched and eroded soils (C1); (ii) mulched and deposited soils (with some samples of both mulched and deposited soils, as well as mulched and eroded soils, C2); and (iii) some samples of both mulched and deposited soils, as well as mulched and eroded soils, C3) (Figure 4b).

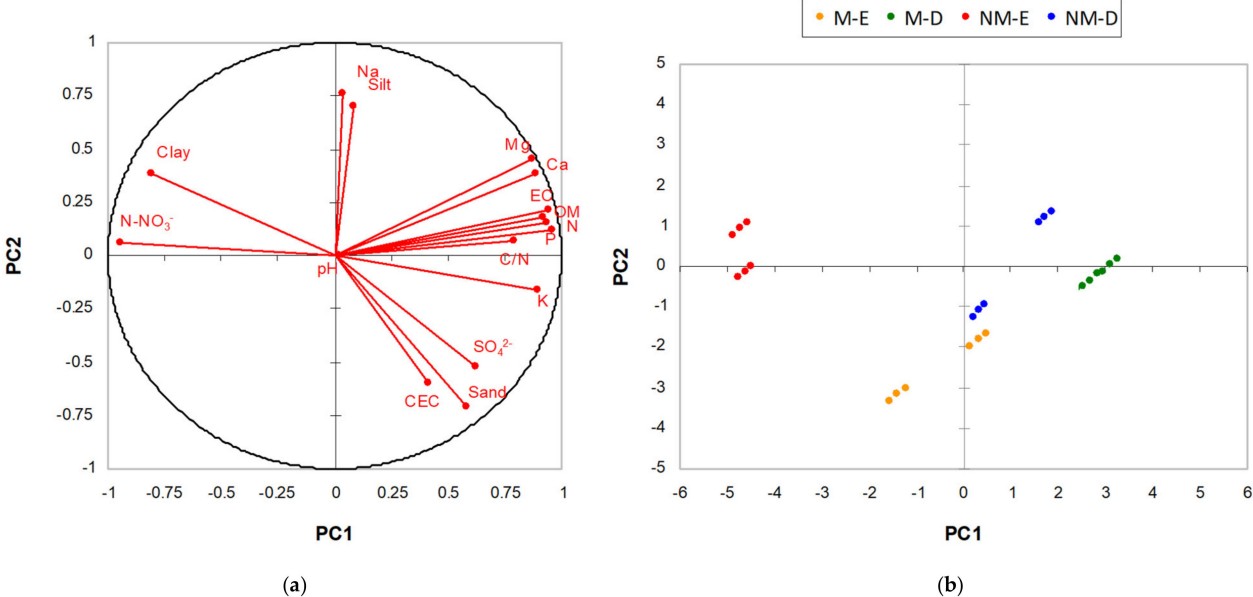

(a)　　　　　　　　　　　　　　(b)

**Figure 4.** Loadings of the soil properties (**a**) and scores of the soil samples (**b**) on the first two principal components (PC1 and PC2) provided by the PCA based on samples of eroded and deposited sediments of soils affected by a wildfire and treated with straw mulch in Liétor (Castilla-La Mancha, Spain). Legend: M-E = mulched and eroded soil; M-D = mulched and deposited soil; NM-E = non-mulched and eroded soil; NM-D = non-mulched and deposited soil.

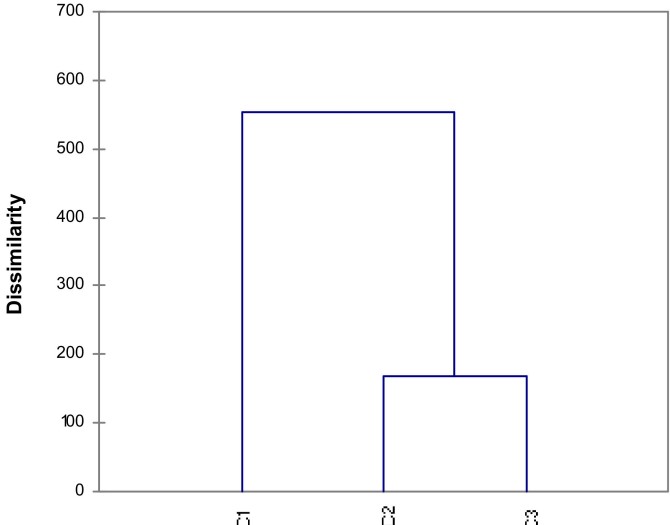

**Figure 5.** Dendrogram provided by the agglomerative hierarchical cluster analysis (AHCA) based on samples of eroded and deposited sediments of soils affected by a wildfire and treated with straw mulch in Liétor (Castilla-La Mancha, Spain). Legend: C1, C2, and C3 are the clusters of soil samples identified by AHCA; the dashed line is the level of similarity. C1 = All NM-E plots; C2 = NM-S + M-E plots; C3 = NM-S + M-E + M-S plots.

## 4. Discussion

Despite its erosive nature (rainfall height over 13 mm, according to Wischmeier and Smith [51]), the first post-fire event produced a limited runoff (less than 1 mm) and erosion (less than 0.006 tons/ha) in burned soil. Increases in surface runoff and soil erosion after wildfires in forests have been widely reported (e.g., [10,14,52]). Mulching reduced the soil's hydrological response by more than 20% (in terms of runoff) and 60% (in terms of soil loss). According to Smets et al. [53], the reduction in runoff is due to higher water infiltration, while soil erosion lowers due to both decreased splash erosion and increased resistance

to flow. As a matter of fact, mulching provides cover of soil surface, reduces the kinetic energy of raindrops impacting on soil, increases its hydraulic conductivity, and creates an obstacle to overland flow, with reduced detachment particles [42,54–56]. The mulching effectiveness measured in our study is lower compared to some other studies (reductions in runoff by 41% in Prats et al. [57,58], and by 57% in Prats et al. [59]; and in soil losses by about 90% in Shakesby et al. [60] and Prats et al. [57–59]. However, this variability in mulching effectiveness can be justified by the different soil and vegetation characteristics, as well as the doses of mulch material distributed over ground (2 tons/ha in our study against amounts up to 10–12 tons/ha in the experiences of Prats et al. [57–59]).

The changes in soil quality detected after wildfire and the monitored rainfall event were in general significant for almost all the analyzed properties. Erosion did not alter soil pH, CEC, and contents of the medium and coarser fractions of mobilized sediments, but determined an enrichment in clay particles, an increase in salinity (shown by the higher EC), and a depletion in OM, nutrients, nitrates, and micro-elements, which were transported in the sediment flows and deposited downstream of the eroded areas. Several studies have shown significant changes in OM and nutrient contents in soils after wildfire compared to unburned soils (e.g., [2,61,62]). Decreases in OM [26,63] and nutrients [64] are somewhat expected after high-severity fires, since substantial consumption of organic matter begins at temperatures between 200 and 250 °C [11,65], and nitrogen is partially lost due to volatilisation [66], while the other part changes its form [65]. In the mass balance, the OM and nutrients lost by combustion or volatilisation are higher compared to the amounts released by ash and incorporated into the soil due to leaching [26,67,68]. In addition, the concentrations of cations ($Ca^{2+}$, $Mg^{2+}$, and $K^+$), and the anion $SO^{2+}_4$ are known to noticeably increase after a wildfire [65,69].

Mulching played important effects on soil quality after the monitored event. For some parameters, this post-fire management technique allowed limiting the magnitude of these changes in the sediments from mulched and eroded areas compared to the untreated soils. The eroded fraction of clay was lower by 40% and nitrate transport decreased by about 80%. In contrast, large increases in OM (more than 120%), nutrients (up to 220% for phosphorous), micro-elements (mainly magnesium, +60%), and electrical conductivity (+44%) were surveyed in the eroded and mulched areas. This result may be due to the low runoff and erosion rates detected in mulched areas, which reduced the loss of soil particles (on which some compounds are adsorbed) due to both rainsplash erosion and overland flow. With specific regard to OM—which is the most important component in ecosystem dynamics [70]—according to Prats et al. [59], the application of mulch cover leads to an increase in the upper soil layer from 40 to 120%, due to the high OM content of forest residues (up to 90%), depending on both the type of the soil and the mulch application rate [59]. This enrichment of the organic matter content at the soil surface is thought to be a key factor for supporting fertility, productivity, and carbon fixation in mulched soils [42,71,72]. However, verification should be carried out on the fate of nitrates, the incorporation of which in soil could represent hazardous groundwater pollution. The reduction in this parameter noticed in both sediments eroded and deposited in the mulched areas compared to the untreated zone seems to confirm this risk.

The variability of all the analyzed parameters in the deposited sediments from mulched and non-mulched areas was lower than 10–15%. Presumably, these elements and compounds were transported downstream by the overland flow, since the only noticeable variations measured in the deposited sediments of mulched areas were detected for sulphates (about +120%) and nitrates (−99%), and, by a much lesser extent (+40%) for phosphorus. The downstream transport of the latter element should be considered with caution, since an increase in its concentration in valley water bodies (mainly lakes and reservoirs) could lead to the eutrophication risk with damage to plants and animals [73].

This mass balance means that mulching is able to limit runoff and erosion rates in burned areas, and this technique cannot induce land degradation in burned areas, which avoid reduction in soil quality and increase in transport of polluting compounds

downstream of the wildfire-affected forest areas. In contrast, the higher quality of mulched areas is evident, thanks to the higher levels of OM, nutrients, and micro-elements, which should support soil fertility and plant regeneration.

The erosion/deposition dynamics were different between the mulched and non-mulched areas, as shown by the PCA and AHCA. This was shown by the first principal component, which was associated with contents of OM, all nutrients, Ca, Mg, and EC, and discriminated mulched and eroded soils, which were affected by large changes in these soil properties compared to the other soil conditions, which instead were subject to a lower variability.

## 5. Conclusions

Compared to burned and untreated areas, the physico-chemical characterisation of mulched soils subject to erosion due to the first intense rainfall event after a wildfire in a Mediterranean pine forest has shown:

- A reduction in runoff ($-20\%$) and in soil erosion ($-60\%$) thanks to mulch cover;
- Significant changes in several properties of burned and untreated surface soils (increase in salinity; and reductions in OM, nutrients, nitrates and micro-elements);
- Effectiveness of mulching on the overall soil quality of eroded areas (large increases in OM, nutrients, and micro-elements);
- Transport of some compounds downstream of the fire-affected areas (mulched or not), although the mobilized amounts were quite low.

Phosphorous runoff towards valley areas and nitrate incorporation into the soil, detected in both mulched and untreated areas, require attention, since these processes may cause eutrophication of water bodies or nitrate pollution in groundwater.

Overall, the study confirmed the working hypothesis that mulching is effective in the short-term in limiting the lost in organic matter, nutrient, and micro-element contents compared to untreated soils, and does not alter soil quality, with beneficial effects on soil fertility and plant regeneration in wildfire-affected areas. Ongoing studies continuing this study should support the preliminary results of this experimental investigation, carrying out a complete mass balance—consisting of water fluxes in addition to the eroded and deposited sediments—exploring the effects of other post-fire management techniques (e.g., log erosion barriers, contour-felled log debris, etc.) and validating these findings in other environmental contexts.

**Author Contributions:** M.E.L.-B. and D.A.Z.: conceptualization, methodology, and formal analysis. M.E.L.-B., D.A.Z. and M.P.: writing—original draft preparation. All authors have read and agreed to the published version of the manuscript.

**Funding:** This research received no external funding.

**Institutional Review Board Statement:** Not applicable.

**Informed Consent Statement:** Not applicable.

**Data Availability Statement:** The data presented in this study are available upon request from the corresponding author.

**Conflicts of Interest:** The authors declare no conflict of interest.

**Abbreviations:** M-E = mulched and eroded soil; M-D = mulched and deposited soil; NM-E = non-mulched and eroded soil; NM-D = non-mulched and deposited soil; EC = electrical conductivity; OM = organic matter content; N = total nitrogen content; P = total phosphorous content; K = potassium content; Na = sodium content; Ca = calcium content; Mg = magnesium content; $N\text{-}NO_3^-$ = nitrate content; $SO_4^{2-}$ = sulphate content; C/N = carbon/nitrogen ratio; CEC = cation exchange capacity.

# Appendix A

**Table A1.** Values of properties of eroded and deposited sediments of soils affected by a wildfire and treated with straw mulch in Liétor (Castilla-La Mancha, Spain).

| Soil Properties | Soil Condition/Erosion Dynamics | | | | | | | |
|---|---|---|---|---|---|---|---|---|
| | Mulched/Erosion | | Mulched/Deposition | | Non-Mulched/Erosion | | Non-Mulched/Deposition | |
| | Mean | Std. Dev. | Mean | Std. Dev. | Mean | Std. Dev. | Mean | Std. Dev. |
| Sand (%) | 60.23 | 2.31 | 59.08 | 1.24 | 52.70 | 2.07 | 58.01 | 2.07 |
| Silt (%) | 32.29 | 2.31 | 32.38 | 2.10 | 33.52 | 0.78 | 33.44 | 2.96 |
| Clay (%) | 8.93 | 0.29 | 9.99 | 0.95 | 15.23 | 1.43 | 10.00 | 0.95 |
| pH (-) | 8.62 | 0.08 | 8.58 | 0.09 | 8.58 | 0.09 | 8.54 | 0.09 |
| EC (mS/cm) | 0.28 | 0.05 | 0.35 | 0.01 | 0.19 | 0.00 | 0.37 | 0.02 |
| OM (%) | 11.01 | 1.78 | 19.53 | 3.69 | 4.92 | 0.19 | 18.91 | 1.98 |
| N (%) | 0.45 | 0.06 | 0.61 | 0.06 | 0.27 | 0.05 | 0.57 | 0.10 |
| P (%) | 35.62 | 12.27 | 67.10 | 2.60 | 10.86 | 0.95 | 48.51 | 12.74 |
| K (%) | 2.38 | 0.21 | 2.32 | 0.16 | 1.25 | 0.09 | 2.13 | 0.29 |
| Na (%) | 0.30 | 0.07 | 0.29 | 0.06 | 0.34 | 0.04 | 0.33 | 0.06 |
| Ca (%) | 43.73 | 4.05 | 55.33 | 1.93 | 39.85 | 0.48 | 52.01 | 6.14 |
| Mg (%) | 6.39 | 1.98 | 8.89 | 0.25 | 3.97 | 0.32 | 10.24 | 2.78 |
| $N\text{-}NO_3^-$ (%) | 3.11 | 2.52 | 0.02 | 0.02 | 14.94 | 3.39 | 1.46 | 1.18 |
| $SO_4^{2-}$ (%) | 35.20 | 6.85 | 37.15 | 0.87 | 13.20 | 3.74 | 16.40 | 0.21 |
| C/N (-) | 14.41 | 1.57 | 18.81 | 2.61 | 11.25 | 1.15 | 19.91 | 1.62 |
| CEC (meq/100 g) | 22.51 | 1.87 | 20.84 | 0.60 | 18.46 | 0.35 | 19.05 | 1.31 |

Note: see the list of abbreviations (Abbreviations) for acronym explanations.

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
