# Peer review of "Short-Term Changes in Erosion Dynamics and Quality of Soils Affected by a Wildfire and Mulched with Straw in a Mediterranean Forest"

_soilsystems, doi:10.3390/soilsystems5030040_

Round 1

Reviewer 1 Report

The communication under review, “Short-term changes in erosion dynamics and quality of soils affected by a wildfire and mulched with straw in a Mediterranean forest” presents a beginning of a worthy effort on a particularly interesting topic. Data are sound and evidence seems to support the conclusions. This communication is nicely written, although it will benefit from some English revision. Additionally, the authors should address several items, both major and minor, before this article is published.

Fig 1: please insert scale to the aerial photo. Study site size is unclear.

Chapter 2.3.1: please improve the sample collection methods. How many water/sediment samples did you collect? How much of it did you analyze? Was it the whole soil amount collected or only part of it (sample)? Please see my comments re sampling method in the text.

Fig 3: do the bars express average values?  What are a and b panels? Please explain in the caption to figure 3.

Figure 4 is missing.

Please see minor corrections and remarks within the attached original text.

Overall, I enjoyed reading the manuscript and learned from it. Thank you!

Author Response

Reviewer # 1

Comment

The communication under review, “Short-term changes in erosion dynamics and quality of soils affected by a wildfire and mulched with straw in a Mediterranean forest” presents a beginning of a worthy effort on a particularly interesting topic. Data are sound and evidence seems to support the conclusions. This communication is nicely written, although it will benefit from some English revision. Additionally, the authors should address several items, both major and minor, before this article is published.

Reply

Dear Prof./Dr.,

thanks a lot for your revision work that we have considered very useful to improve our MS. You will find below our replies to all your comments. However, we address You to the file containing the revised paper and attached to the resubmission.

Comment

Fig 1: please insert scale to the aerial photo. Study site size is unclear.

Reply

Done.

Comment

Chapter 2.3.1: please improve the sample collection methods. How many water/sediment samples did you collect? How much of it did you analyze? Was it the whole soil amount collected or only part of it (sample)? Please see my comments re sampling method in the text.

Reply

Thanks for your comment. The following info has been added: “Overall, twelve runoff samples and twelve soil samples, both collected in tanks and sediment fences, respectively, were analyzed. For soil samples, 500 g were oven dried (at 105 °C) for 24 hours in the laboratory” (see lines 169-170 of the revised MS with tracks).

Comment

Fig 3: do the bars express average values?  What are a and b panels? Please explain in the caption to figure 3.

Reply

The bars express average values. We have completed the caption for a better understanding.

Comment

Figure 4 is missing.

Reply

Sorry for the mistake, due to a formatting problem. We have added the missing figure.

Comment

Please see minor corrections and remarks within the attached original text.

Reply

Thanks a lot for all Your suggestions and comments. Please refer to the revised MS with tracks, where we have highlighted all the corrections made accordingly to Your proposals.

Comment

Overall, I enjoyed reading the manuscript and learned from it. Thank you!

Reply

Thanks a lot again for Your opinion!

Reviewer 2 Report

The methods of Hydrological Observations is not informative enough, and needs the sample plots’ info. (e.g. terrain slope and vegetation coverage). It is also suggested to supplement the layout of sample plots and observation facilities.

The amount of data is far from enough. There is only the runoff and sediment observation data of one rainfall event. It is suggested to supplement the observation data of different rainfall event.

Figure 2 needs to be redone. Please use double ordinate axis. Different letters a and b should be placed above the data axis.

The meanings of a and b in the Figure 3 need to be added in the title.

Line 203-204 and line 215-216 of Page 6, Figure 3a and figure 3b are misplaced.

Author Response

Reviewer # 2

Dear Prof./Dr.,

thanks a lot for your revision work that we have considered very useful to improve our MS. You will find below our replies to all your comments. However, we address You to the file containing the revised paper and attached to the resubmission.

Comment

Methods of Hydrological Observations is not informative enough, and needs the sample plots’ info. (e.g. terrain slope and vegetation coverage). It is also suggested to supplement the layout of sample plots and observation facilities.

Reply

Information about experimental sites and plots was added to the text. “Aspect was north and slope was 30-35% for all plots. The vegetation type before wildfire was the same for all plots (see Lucas-Borja et al., 2020a for more details)” (see lines 163-164 of the revised MS with tracks).

Comment

The amount of data is far from enough. There is only the runoff and sediment observation data of one rainfall event. It is suggested to supplement the observation data of different rainfall event.

Reply

This important observation sources from a lack of clearness of the original MS. We apologize for this. Our intention for this short communication was the evaluation of the changes in the properties of the soil (mulched or not subject to any post-fire management technique) eroded in the early stage of the so-called ‘window of disturbance’ that occurs immediately after wildfire. Although this window of disturbance lasts some months, the first event, in particular, is responsible for the main changes in soil properties and the highest erosion. This is the reason why we have only focused the first event rather than studying the following rainstorms, when the disturbance is lower and decreasing. About this statement, ample literature exists (e.g., Lucas-Borja et al., 2019-Science of the Total Environment; Lopes et al., 2020-Cuadernos de Investigacion geografica; Zituni et al., 2019-International Journal of Wildland Fire). Moreover, in another experimental site (Southern Italy, in the same climatic environment), we are monitoring the entire post-fire period in a pine stand, to confirm the findings of these statements (major influence of the first event on the total annual amounts of runoff and erosion). Unfortunately, the data are under review, therefore we can not share. We add only a figure, which may support this statement (Legend: U = unburned soil; B = burned soil; B+M = burned and mulched soil). We hope that this may be convincing. We have relevant information in the revised MS (see lines 101-102).

Comment

Figure 2 needs to be redone. Please use double ordinate axis. Different letters a and b should be placed above the data axis.

Reply

Done.

Comment

The meanings of a and b in the Figure 3 need to be added in the title.

Reply

Done.

Comment

Line 203-204 and line 215-216 of Page 6, Figure 3a and figure 3b are misplaced.

Reply

Corrected in both occurrences.

Author Response

Reviewer # 3

Comment

Dear Authors! I with interest read your Communication entitled: “Short-term changes in erosion dynamics and quality of soils affected by a wildfire and mulched with straw in a Mediterranean forest”.

Due to climate change the frequency of wildfires increases worldwide. The wildfires lead to many ecological and economic problems. Wildfires release large amounts of carbon dioxide, they also lead to a deterioration of the air quality, and loss of property, crops, resources, animals and people. The research of Authors has a high actuality and topic fits to Soil Systems journal.

The study of mulch effect on soil erosion at burned forest areas is interesting, while it less studied. Usually in the forests sites there is no soil erosion/runoff (or is minimal, or rates of soil formation exceeds it) due to: developed vegetation, good water permeability (infiltration), trees decrease the kinetic energy of drops during rainfall, thus decreasing the splash erosion. After wildfire such properties of soil is gradually decreases and soil became vulnerable to soil erosion.

So the effect of mulching treatments on soil affected by wildfires will be interesting for readers.

Reply

Dear Prof./Dr.,

thanks a lot for your revision work that we have considered very useful to improve our MS. You will find below our replies to all your comments. However, we address You to the file containing the revised paper and attached to the resubmission.

Comment

At almost the Communication is well written, however, I have some comments and suggestions:

  • In abstract please add the concrete mulching dose, precipitation amount of rainfall event and erosion/runoff rates and dates when wildfire and research was conducted.

Reply

Done.

Comment

  • In Keywords “Window of disturbance” could be omitted.

Reply

Removed.

Comment

  • L. 37 and in all text, please check the journal rules, it seems that the References and cited literature should be prepared as [1, 2] not (Marzaioli et al., 2010; Lucas-Borja et al., 2020a).

Reply

Checked and changed everywhere in the text.

Comment

  • Figure 1. If put the mark of “North” and add the coordinates net it will be more decent.

Reply

Done. North bar has been added and the same for the coordinates (38.5164048N, -1.8318104E).

Comment

  • L. 133. Please add the year of study.

Reply

Added.

Comment

  • L. 139. I didn’t found the “above-cited classification”.

Reply

Removed.

Comment

  • L .140. Please add the year of study and describe the composition of mulch.

Reply

Added.

Comment

  • L. 146. Please add the year.

Reply

Added.

Comment

  • L. 156. The additional description is need. How you measured the total runoff volume? You used a collection tanks? Which volume?

Reply

The total amount of generated runoff was stored in tanks of 100 L. This information has been added to the text (see line 161 of the revised MS with tracks).

Comment

  • L. 190. Additionally here you could show the rates of soil erosion in (t/ha), it more informative. Also you could use the units (t/ha) in Figure 2.

Reply

We have added also the rates in tons/ha in the text, but not in the figure to avoid confusion.

Comment

  • L. 201. Please describe in the text that the difference between “eroded and deposited” sediments.

Reply

We meant “Soil properties between the eroded (soil remaining on the hillslope after the rainy event) and soil deposited in sediment fences”. Information added (see lines 234-235).

Comment

  • Figure 3 (a) for which soil layer? It is very important.

Reply

We are referring to soil surface. Information added.

Comment

  • L. 204. It should be Figure 3b?

Reply

Corrected.

Comment

  • L. 215. For which soil layer?

Reply

We are referring to soil surface.

Comment

  • L.309-312. Please add the reference for such claim.

Reply

Reference added.

Comment

  • L. 331. For which soil layer?

Reply

We are referring to soil surface.

Comment

In Introduction it useful if you will be mentions about frequency of wildfires in Spain and how many areas of forest was affected by fires. Also interesting to compare the soil erosion rates of forests affected by wildfires of your studied site with other regions/countries. For example you could use the following papers (https://doi.org/10.1016/j.jenvman.2020.110491,  https://doi.org/10.1134/S1064229319040070, https://doi.org/10.1002/hyp.13932).

Reply

Thank you very much for this suggestion. References and information added.

Comment

Wish good luck in your present and future research. With best regards, …..

Reply

Thank a lot for your constructive review.

Round 2

Reviewer 2 Report

The manuscript has been greatly improved and is worthy of publication.